# Impact of [^18^F]FDG PET/CT in the Assessment of Immunotherapy-Induced Arterial Wall Inflammation in Melanoma Patients Receiving Immune Checkpoint Inhibitors

**DOI:** 10.3390/diagnostics13091617

**Published:** 2023-05-03

**Authors:** Shaghayegh Ranjbar, Seyed Rasoul Zakavi, Roya Eisazadeh, Seyed Ali Mirshahvalad, Julia Pilz, Zahra Jamshidi-Araghi, Gregor Schweighofer-Zwink, Peter Koelblinger, Christian Pirich, Mohsen Beheshti

**Affiliations:** 1Division of Molecular Imaging & Theranostics, Department of Nuclear Medicine, University Hospital Salzburg, Paracelsus Medical University, 5020 Salzburg, Austriaeisazadeh.roya@gmail.com (R.E.); mirshahvalad.sa@gmail.com (S.A.M.); julia.pilz@ordensklinikum.at (J.P.); g.schweighofer-zwink@salk.at (G.S.-Z.); c.pirich@salk.at (C.P.); 2Nuclear Medicine Research Center, Mashhad University of Medical Sciences, Mashhad 13944-91388, Iran; 3Joint Department of Medical Imaging, Toronto General Hospital, University Health Network, University of Toronto, Toronto, ON M5G 2C4, Canada; 4Department of Urology, Ordensklinikum Linz, 4020 Linz, Austria; 5Department of Dermatology, University Hospital, Paracelsus Medical University, 5020 Salzburg, Austria

**Keywords:** FDG, PET/CT, immunotherapy, vasculitis, melanoma

## Abstract

We aimed to investigate the role of [^18^F]FDG positron emission tomography/computed tomography (PET/CT) in the early detection of arterial wall inflammation (AWI) in melanoma patients receiving immune checkpoint inhibitors (ICIs). Our retrospective study enrolled 95 melanoma patients who had received ICIs. Inclusion criteria were ICI therapy for at least six months and at least three [^18^F]FDG PET/CTs, including one pretreatment session plus two scans three and six months after treatment initiation. AWI was assessed using quantitative and qualitative methods in the subclavian artery, thoracic aorta, and abdominal aorta. We found three patients with AWI visual suspicion in the baseline scan, which increased to five in the second and twelve in the third session. Most of these patients’ treatments were terminated due to either immune-related adverse events (irAEs) or disease progression. In the overall population, the ratio of arterial-wall maximum standardized uptake value (SUVmax)/liver-SUVmax was significantly higher three months after treatment than the pretreatment scan in the thoracic aorta (0.83 ± 0.12 vs. 0.79 ± 0.10; *p*-value = 0.01) and subclavian artery (0.67 ± 0.13 vs. 0.63 ± 0.12; *p*-value = 0.01), and it remained steady in the six-month follow-up. None of our patients were diagnosed with definite clinical vasculitis on the dermatology follow-up reports. To conclude, our study showed [^18^F]FDG PET/CT’s potential to visualise immunotherapy-induced subclinical inflammation in large vessels. This may lead to more accurate prediction of irAEs and better patient management.

## 1. Introduction

Melanoma is a form of cancer that mostly arises from cutaneous melanocytes, though it also presents in other locations such as leptomeninges or uvea [1]. Surgery is the treatment of choice when the tumour is localised or when there is a regional disease or a single metastasis (AJCC stages I/II as well as resectable stage III). However, systemic therapy is mandatory in high-risk patients (stage IV and unresectable stage III) [2,3]. In advanced melanoma (unresectable stages III and IV), immunotherapy with immune checkpoint inhibitors (ICIs) or targeted therapy with B-RAF/MEK inhibitors (targeted therapy) is used as first-line systemic therapy [2,4,5]. Immunotherapy is independent of mutation status and could cover a broader spectrum of patients [4,5,6]. Cytotoxic T lymphocyte-associated antigen 4 (CTLA-4) monoclonal antibodies and programmed death-1 (PD-1) inhibitors are the two major types of ICIs currently in use [7]. Ipilimumab is an FDA-approved monoclonal antibody against CTLA-4. Nivolumab and Pembrolizumab are FDA-approved PD-1 inhibitors for treating unresectable or metastatic melanoma in the form of monotherapy or in combination with Ipilimumab [8,9,10,11,12].

Although the advent of immunotherapeutics has brought dramatic changes in melanoma patients in terms of progression-free (PFS) and overall (OS) survival [13], immune-related adverse events (irAEs) occur at the varying frequency with the use of ICIs since the enhancement of the immune system for cancer prevention may lead to incidental autoimmune-mediated complications possibly affecting multiple organs [13,14,15]. An example of irAEs is immunotherapy-induced large vessel vasculitis, which is either limited to one organ or occurs as a systemic disease [16,17]. The prevalence of immunotherapy-related vasculitis-induced irAEs (e.g., cutaneous purpura, neuropathy, and visceral vasculitis) is approximately 12% [18]. The median interval between the occurrence of vasculitis and the initiation of immunotherapy has been reported to be three months, ranging from as early as eight days to a long period of more than one year [16,19]. Another related event is atherosclerosis, which is a chronic, low-grade inflammatory disease of the arterial wall and a risk factor for myocardial infarction and stroke [20]. Arterial wall inflammation (AWI) is strongly correlated to the risk of atherosclerotic plaque rupture. Additionally, late diagnosis of AWI in the aorta can lead to irreversible vascular dissection and aneurysm or the development of ischemic heart disease, which can have fatal outcomes for patients [21,22].

[^18^F]FDG PET/CT is a valuable method by which to estimate prognosis and predict response to therapy in patients undergoing immunotherapy. It is also helpful for the patients’ follow-up [23,24,25]. [^18^F]FDG PET/CT can also reveal inflammation of the vessel walls. It can determine the extent and severity of large vessel vasculitis as well [26,27,28]. Because evolving subclinical AWI, particularly in the aorta and main branches, could be associated with cardiovascular complications for patients and lead to morbidity and mortality, early detection of inflammatory changes in [^18^F]FDG PET/CT could provide important information to help clinicians to balance the risks and benefits of immunotherapy. Therefore, this study’s main objective is to investigate the role of [^18^F]FDG PET/CT in the early detection of AWI in patients receiving ICIs for the treatment of advanced melanoma and also assess the trend of AWI during immunotherapy.

## 2. Material and Methods

### 2.1. Patient Selection

Overall, 95 patients with advanced cutaneous melanoma who had received anti-PD1 or combination therapy by using anti-PD1 plus CTLA-4 agents immunotherapy for at least 6 months and had undergone at least three [^18^F]FDG PET/CT examinations, including one pretreatment scan (no more than eight weeks before starting immunotherapy) plus two scans three and six months after treatment initiation, were included. The treatment was conducted according to ESMO guidelines, and all patients received systemic therapy as their first-line treatment [29].

All image data were reviewed during treatment. Patients who did not undergo [^18^F]FDG PET/CT before treatment were excluded. Additionally, patients whose images were of poor quality or were not evaluable because of high uptake of brown fat (which affects the estimation of vessel wall uptake) or high burden of liver or mediastinal metastases (which affects the assessment of background uptake) were excluded. None of the patients had a high calcium score or was known for major cardiac events at baseline. Treatment and follow-up for all patients were conducted by the dermatology department. There was no evidence of clinical signs and symptoms of vasculitis in the patients’ dermatology follow-up reports.

### 2.2. Imaging Protocol

All subjects underwent whole-body [^18^F]FDG-PET/CT imaging 60 min after injecting 4.0 MBq/kg [^18^F]FDG intravenously. Each scan was obtained using the same protocol, and imaging was performed on hybrid PET/CT scanners with a comparable spatial resolution (Siemens 923/Biograph 64 mCT (Siemens Healthineers AG, Chicago, IL, USA); Philips Ingenuity TF/Gemini TF 16 (Philips Medical Systems, Andover, MA, USA)). Low-dose CT imaging was performed for attenuation correction and anatomic correlation. PET scans were corrected to account for scattering, attenuation, random coincidences, and scanner dead time.

### 2.3. [^18^F]FDG-PET/CT Visual Assessment

We analysed images using visual and semi-quantitative approaches to assess large vessel wall inflammation. Three vascular regions were evaluated for these analyses, including the subclavian arteries, thoracic aorta, and abdominal aorta. For visual assessment, we considered the uptake of subclavian arteries when it was visible bilaterally along the arterial walls in the maximal intensity projection (MIP) and coronal views. Concerning the aorta, we assessed the thoracic and abdominal aorta walls in the sagittal view and evaluated whether a tramline pattern was present along the arterial wall.

A scoring system was applied for visual assessment for a reproducible data analysis [30]. Four-scale scoring (0–3) was given based on the background activity of the mediastinum and liver as follows:-Grade 0: No vascular uptake (≤mediastinum);-Grade 1: Vascular uptake < liver uptake;-Grade 2: Vascular uptake = liver uptake;-Grade 3: Vascular uptake > liver uptake.

According to this grading method, we categorised the scores of zero and one as negative, two as suspected for vasculitis, and three as definitely positive for vasculitis, respectively, for each vascular region.

The visual method was tested on several patients in a session with three nuclear medicine specialists to standardise the interpretation between the three specialists. Then, the patients were divided into two groups, and two nuclear medicine physicians assessed each group. If there were equivocal findings in visual assessment, the case was discussed with the second physician, and if consensus could not be reached, the issue was discussed with the third specialist to reach an agreement.

### 2.4. [^18^F]FDG-PET/CT Semi-Quantitative Analysis

For the semi-quantitative approach, we manually drew an adjusted volume of interest (VOI) around the following structures in the transverse view: right or left subclavian artery (the one with the higher SUV); thoracic and abdominal aorta. The VOIs were drawn on a slice where we could get the best VOI to include the arterial wall clearly and as far as possible. Areas of vascular calcification and adjacent extravascular activity were excluded.

We checked the exact position of VOIs in sagittal and coronal views to ensure that no extravascular activity was included in the VOI to interfere with the accurate estimation of SUVmax. The SUVmax and SUVmean were measured for the mentioned arteries. The liver and venous blood pool were chosen as reference organs and matched defined VOIs on the superior vena cava (1 cm in transaxial diameter) as the blood pool and on the right lobe of the liver (3 cm in transaxial diameter), where no liver metastases were present. Finally, the target-to-background ratios (the ratio of SUVmax and SUVmean of the vessels to the SUVmax and SUVmean of the liver and blood pool) were calculated and recorded for all vascular regions. All these data were recorded for pretreatment [^18^F]FDG PET/CT scan and follow-up scans throughout the period in which patients went through different cycles of immunotherapy, and each element’s alteration trend was evaluated.

### 2.5. Statistical Analysis

Normality tests were performed for all variables. Since the data were not normally distributed, the Friedman test was used to compare semi-quantitative variables between three sets of images. The Wilcoxon signed ranked test was used for comparison between groups. Visual assessment was performed using the McNemar test. All statistical analyses were performed by IBM SPSS statistics 27.0.1. A two-sided *p*-value < 0.05 was considered statistically significant in all comparisons.

## 3. Results

### 3.1. Patients

Overall, 95 patients were included (female = 45, male = 50; mean age = 60 ± 13.1 years; mean BMI = 26.45 ± 4.67). Detailed characteristics are shown in Table 1. Three kinds of ICIs or a combination of them were used in these patients (Ipilimumab-only = 0, Nivolumab-only = 55, Pembrolizumab-only = 12, Ipilimumab + Nivolumab = 2, Ipilimumab + Nivolumab continue with Nivolumab = 26). Indeed, 67 patients received monotherapy and 28 patients received combination therapy.

We also followed patients after their third [^18^F]-FDG PET/CT session (at 6 months) with a median follow-up period of 383 days (110–948 days). During the follow-up period, 31 (33%) patients finished it because of response to treatment and achieving a favourable response to therapy (according to immunotherapy-modified Positron Emission Tomography Response Criteria in Solid Tumors [imPERCIST]), 27 (28%) patients suspended immunotherapy because of the disease progression, 17 (18%) patients stopped it due to major side effects, 13 (14%) patients had ongoing ICI treatment, 5 (5%) patients stopped treatment due to completion of a planned course of immunotherapy, and 2 (2%) patients discontinued therapy course based on personal request. The reported adverse events included (some patients revealed two events coincidentally) colitis (*n* = 4), pancreatitis (*n* = 4), arthritis (*n* = 3), neuritis (*n* = 3), hypophysitis (*n* = 3), pneumonitis (*n* = 2), bullous pemphigoid (*n* = 2), hepatitis (*n* = 2), myocarditis (*n* = 1), pan uveitis (*n* = 1), exanthem (*n* = 1), lipase elevation (*n* = 1), and eosinophilia (*n* = 1). Six (6%) patients deceased during the follow-up period. In general, our patients revealed complete remission in 33% of the cases during a mean follow-up period of 383 days.

### 3.2. Visual Analysis

In the pre-treatment [^18^F]FDG PET/CT, three patients had grade 2 uptake in the thoracic aorta, while only one had a grade 2 uptake in the abdominal aorta and subclavian artery. Three months after treatment, grade 2 uptake was increased in all vascular territories (Figure 1). The thoracic aorta showed the highest amount of grade 2 uptake in our study in all [^18^F]FDG PET/CT images and was considered for statistical analysis. While only three patients had a grade 2 uptake in the initial [^18^F]FDG PET/CT (PET1), it was increased to five after three months and further increased to twelve patients (seven new cases) six months after immunotherapy. Although the rise was statistically insignificant three months after therapy, it became significant at six months (*p*-value = 0.01). Interestingly, in those seven new patients with visual presentation of inflammation in the third scan, five presented with other irAEs in the follow-up, including colitis, neuritis, hypophysitis, pneumonitis, and bullous pemphigoid, resulting in therapy termination. One patient discontinued therapy due to a personal request. Moreover, three of those five patients with grade 2 uptake in both PET2 and PET3 showed early disease progression thereafter, and the treatment was discontinued.

### 3.3. Semi-Quantitative Analysis

Table 2 shows the mean value of the thoracic aorta SUVmax, SUVmean of the venous blood pool, and SUVmax over the liver and the calculated ratios before treatment and three and six months later. The absolute values of SUVmax and SUVmean were not different between the three groups. However, the ratio of thoracic aorta SUVmax to liver SUVmax was significantly higher in PET2 and PET3 compared to PET1 (*p*-value = 0.01). Further post hoc analysis revealed that the ratio was not significantly different between PET2 and PET3 (*p*-value = 0.43). The same data were depicted in Table 3 and Table 4 for the subclavian artery and abdominal aorta, respectively. In a similar pattern to the thoracic aorta, the vessel-to-liver SUVmax ratio was significantly higher in PET2 than PET1 for the subclavian artery (*p*-value = 0.01). However, this significant trend could not be seen for the abdominal aorta. There was no other significant difference between the three PET series regarding SUVmax, SUVmean, and target-to-background ratios.

Noteworthy, the significance status of parameters between PET examinations remained the same after excluding patients (*n* = 10) who received glucocorticoid treatment during their immunotherapy period (any duration in their first six months) due to minor adverse effects. Moreover, no significant difference was found when comparing semi-quantitative parameters between the monotherapy and combined therapy population. Moreover, exclusively in those 12 patients who presented with visual grade 2, the thoracic aorta SUVmax to liver SUVmax ratio was significantly increased in PET3 compared to PET1 (*p*-value = 0.03).

## 4. Discussion

Immunotherapy with ICIs has completely altered the clinical management of metastatic melanoma [2]. However, this therapy’s adverse outcomes are not rare, and the exact pathogenesis is unknown. [^18^F]FDG PET/CT might detect irAEs even in the asymptomatic, subclinical state. Although a change in management is required only in special circumstances, this molecular-level assessment can be helpful in closely monitoring patients and modifying the management if needed, including the initiation of irAE-related treatment or immunotherapy termination, to prevent clinically significant issues later on. One of the important irAEs is cardiovascular events, which may result in a healthcare burden [31]. In this regard, the role of PD-1, PD-L1, and CTLA-4 has been implicated in the pathogenesis of medium- and large-artery vasculitis in recent experimental and genetic research [16,32,33].

We studied 95 patients who received immunotherapy for advanced melanoma and underwent baseline [^18^F]FDG PET/CT and follow-up scans three and six months after immunotherapy. The number of patients visually suspected of vasculitis gradually increased from 3 to 5 to 12 from the first to second to third [^18^F]FDG PET/CT scans. Furthermore, we tried to quantify this molecular-level inflammation using target-to-background ratios. Vessel wall SUVmax to liver SUVmax ratios in the thoracic aorta and subclavian artery in the second [^18^F]FDG PET/CT scan were significantly altered compared to the first one in the whole population. However, the gradual increase seen in the visual assessment was not evident between the second and third [^18^F]FDG PET/CT scans, and the overall mean calculations of semi-quantitative parameters remained steady. After excluding cases who received glucocorticoid treatment, no change was seen in the obtained results. Moreover, no significant difference existed between monotherapy and combination therapy patients. Notably, none of our patients were diagnosed with definite clinical vasculitis in the dermatology follow-up reports.

Previous studies have shown that [^18^F]FDG PET/CT has an increasing role in the diagnosis of large-vessel vasculitis, demonstrating increased uptake in the walls of large-sized arteries [8,9,34]. A systematic review by Daxini et al. demonstrated an association between immunotherapy and vasculitis, which was more frequently seen in large arteries (e.g., GCA-isolated aortitis) and nervous system vessels after one to 15 cycles of treatment with a median occurrence time of three months [19]. Similarly, the patients in our study revealed a gradual increase in the number of suspected subjects for AWI in three-month and six-month follow-ups. Moreover, our whole population developed some quantitative degrees of large vessel inflammation in [^18^F]FDG PET/CT images at three-month follow-up, which also remained after six months. Notably, we found no difference between patients treated with monotherapy versus combination therapy, which was also reported in a previous study by Salem et al. [32].

According to the medical literature, not only can active large-vessel vasculitis be diagnosed with [^18^F]FDG PET/CT with higher sensitivity than other techniques; it can also be used to track the changes during the treatment [25,26,27]. However, no study has been performed on the utility of [^18^F]FDG PET/CT in evaluating immunotherapy-induced vasculitis in melanoma patients, in particular. In this study, we used the interpretation methods published previously in the literature, providing evidence-based recommendations for [^18^F]FDG PET/CT imaging in large vessels [27,28,29,35]. As concluded by Puppo et al. in their systematic review regarding vasculitis assessment via [^18^F]FDG PET/CT, our results may also support the fact that qualitative methods are more specific than semi-quantitative analysis [35]. Interestingly, more than half of our patients who showed grade 2 uptake between three and six months also revealed other irAE presentations, including colitis, neuritis, hypophysitis, pneumonitis, and bullous pemphigoid. This might support the visual findings of [^18^F]FDG PET/CT. Moreover, it should be noted that the majority of our visually suspected patients’ treatments were terminated due to either irAEs or disease progression. Thus, early termination of the therapy might prevent the incidence of clinical vasculitis in the follow-up. Notably, Stellingwerff et al. showed that the visual grading method had the greatest accuracy to support the diagnosis when grade 3 (arterial uptake higher than liver, similar to our definition) was considered vasculitis, with a sensitivity of 92% and a specificity of 91% [36]. However, in the current study, we did not score 3 (definite vasculitis) on visual grading in any of the patients, which would allowed us to assess this threshold.

For the quantitative comparison between successive [^18^F]FDG PET/CT scans, it was mentioned that the use of target-to-background SUV ratio is preferable to absolute SUVs because it limits the error caused by signal quantification, patient weight, amount of radiotracer injected, and radiotracer acquisition time, and is also independent of different PET/CT scanners [35,37]. It was recommended that the ratio between the average of SUVmax of the vessel walls and the SUVmax of one liver region (preferably the right hepatic lobe) or the ratio between the average of SUVmax of the vessel walls and the average of SUVmax of several venous VOIs should be used for semi-quantitative assessment [23], was the case in our study. Some researchers stated that the target-to-blood pool ratio could provide promising results and recommended it for research studies, but they added that the calculated ratio by the liver is also a valuable method [25,26]. We found that absolute SUVs were not significantly different between the three sets of images in general and in our visually suspected patients. However, there was an increase in arterial-to-background calculations. This difference was only prominent in the arterial-to-liver SUVmax, and not in the arterial-to-blood pool ratio. Thus, although liver inflammation may happen in patients under immunotherapy, our results may support the fact that the arterial-to-liver SUVmax ratio can be more reliable than arterial-to-blood pool SUVmax ratios in the interpretation of AWI [26,27,28,38,39]. Noteworthy, contrary to our findings, one study suggested that SUVmean is preferable to SUVmax and is a more sensitive, specific, and repeatable method of representing disease activity than SUVmax [40].

Previous studies have shown that nadir [^18^F]FDG uptake in the arteries correlates with the clinical improvement of patients. However, to what extent [^18^F]FDG PET/CT normalises fully after clinical remission is unclear [21,22,23]. In our study, there was a gradual increase in the number of cases suspected of AWI. Moreover, the average calculations of the semi-quantitative parameters showed that the molecular-level inflammation may become present in a three-month period after initiation of ICIs and remain steady until six months. These findings may reveal the fact that this inflammation is not a short-term problem, at least at the molecular level. This ongoing inflammation may cause other health-related issues, such as atherosclerosis, vasculitis, myocarditis, and polymyalgia rheumatica [31,32,41]. Thus, [^18^F]FDG PET/CT may have a role in detecting patients at risk, which requires further investigation.

To our knowledge, this study was the first evaluation of the role of [^18^F]FDG PET/CT in assessing the molecular signs of immunotherapy-induced vasculitis with a relatively large number of patients. However, due to its retrospective nature, our study was subject to collection bias. Moreover, the relevant laboratory data of the patients (Erythrocyte sedimentation rate [ESR] and C-reactive protein [CRP]) were not evaluated during the treatment period. In addition, in terms of visual assessment, although both nuclear medicine physicians had sufficient experience in PET/CT reading of melanoma patients in immunotherapy, the assessment of AWI in this cohort was somewhat challenging and might be reader dependent. However, an attempt was made to conduct the study with as little error as possible by standardising the visual assessment between physicians in a meeting held before data collection began.

In conclusion, our study showed that clinical vasculitis following immunotherapy is not a common irAE, as we did not find any definite case of vasculitis in 95 patients. However, the visual assessment revealed some suspected cases of AWI, suggesting a trend toward an inflammatory process at the molecular level. Moreover, the arterial wall to liver SUVmax quantitative analysis supported the findings. This may lead to more accurate prediction of irAEs, and improved management of selected high-risk patients with comorbidities. Therefore, [^18^F]FDG PET/CT seems to be a promising tool for decision-making in melanoma patients. However, further studies are needed to assess its role, especially in predicting the risk of atherosclerosis associated with immunotherapy.

## Figures and Tables

**Figure 1 diagnostics-13-01617-f001:**
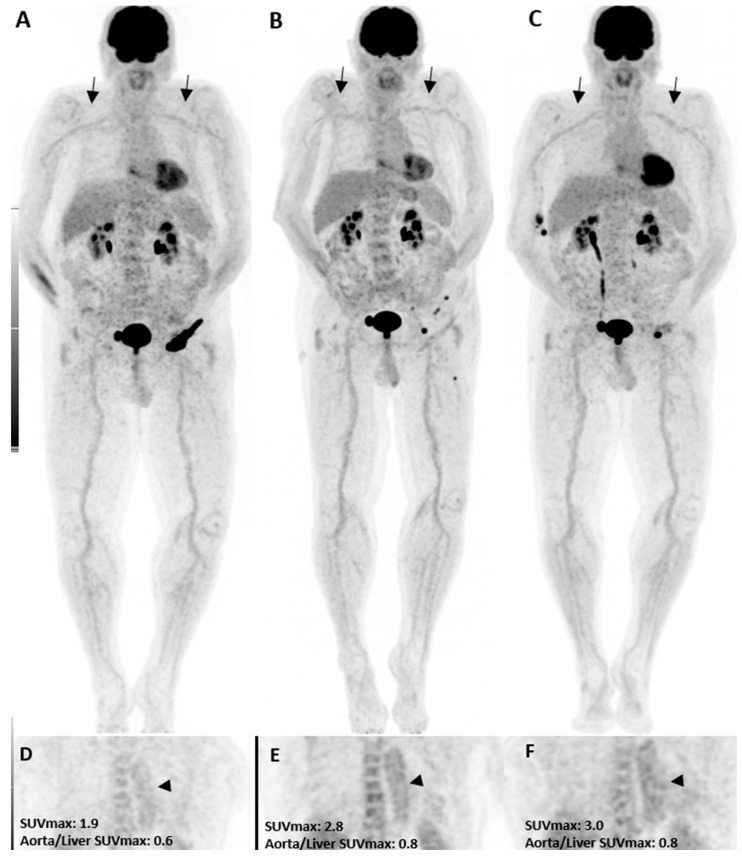
A patient with the diagnosis of melanoma with ongoing immune checkpoint inhibitors treatment without a history of cardiovascular diseases or clinical evidence of vasculitis. [^18^F]FDG PET/CT maximum intensity projections (**B**,**C**), as well as coronal (**E**,**F**) views, three and six months after treatment initiation, are shown. The mild increase in uptake intensity of the aorta and subclavian arterial walls was visually seen compared to the pretreatment (**A**,**D**, arrows) images. Semi-quantitative analyses by SUVmax were correlated with visual findings showing a mild increasing pattern of [^18^F]FDG uptake suggestive of immunotherapy-related mild arterial wall inflammation.

**Table 1 diagnostics-13-01617-t001:** Patient’s Characteristics (*n* = 95).

Mean Age (Years)	60 ± 13.7, Range 21–88
Immune Checkpoint Inhibitors *	
Nivolumab	*n* = 55 (57.9%)
Pembrolizumab	*n* = 12 (12.6%)
Ipilimumab + Nivolumab	*n* = 2 (2.1%)
Ipilimumab + Nivolumab; continue with Nivolumab	*n* = 26 (27.4%)
Treatment Termination Reason/Status	
Undefined	*n* = 6 (6.3%)
Ongoing	*n* = 13 (13.7%)
Response to Treatment	*n* = 31 (32.6%)
Progression	*n* = 27 (28.4%)
Immune-related adverse effects	*n* = 13 (13.7%)
Therapy completion	*n* = 5 (5.3)
AJCC Staging	
IIIA	*n* = 1 (1%)
IIIB	*n* = 11(11.6%)
IIIC	*n* = 1 (1%)
IIID	*n* = 27 (28.4%)
IV	*n* = 54 (56.8%)
Undefined	*n* = 1 (1%)

* Ten patients received glucocorticoid treatment in addition to their regimen. None of the patients received only Ipilimumab as the regimen.

**Table 2 diagnostics-13-01617-t002:** Semi-quantitative analysis in the thoracic aorta.

Semi-Quantitative Variables		PET Session			PET1 vs. PET2 (*p*-Value)	PET2 vs. PET3 (*p*-Value)
		PET1	PET2	PET3		
**Thoracic aorta SUV**	**SUVmax**	**2.58 ± 0.44**	2.63 ± 0.43	2.63 ± 0.46	0.17	0.95
	**SUVmean**	1.74 ± 0.26	1.76 ± 0.26	1.77 ± 0.28	0.47	0.71
**Thoracic aorta SUV/Liver SUVmax**	**SUVmax**	0.79 ± 0.10	0.83 ± 0.12	0.82 ± 0.14	0.01 *	0.43
	**SUVmean**	0.54 ± 0.77	0.55 ± 0.11	0.55 ± 0.08	0.16	0.56
**Thoracic aorta SUV/Blood pool SUVmax**	**SUVmax**	1.23 ± 0.18	1.26 ± 0.19	1.30 ± 0.25	0.23	0.14
	**SUVmean**	0.83 ± 0.11	0.85 ± 0.12	0.87 ± 0.12	0.37	0.09

* statistically significant.

**Table 3 diagnostics-13-01617-t003:** Semi-quantitative analysis of the subclavian artery.

Semi-Quantitative Variable		PET1	PET2	PET3	PET1 vs. PET2 (*p*-Value)	PET2 vs. PET3 (*p*-Value)
**Subclavian artery SUV**	**SUVmax**	2.04 ± 0.36	2.10 ± 0.40	2.07 ± 0.36	0.12	0.46
	**SUVmean**	1.40 ± 0.26	1.43 ± 0.33	1.40 ± 0.24	0.30	0.29
**Subclavian artery SUV/Liver SUVmax**	**SUVmax**	0.63 ± 0.12	0.67 ± 0.13	0.65 ± 0.11	0.01 *	0.15
	**SUVmean**	0.43 ± 0.87	0.45 ± 0.11	0.44 ± 0.09	0.15	0.35
**Subclavian artery SUV/Blood pool SUVmax**	**SUVmax**	0.98 ± 0.17	1.01 ± 0.18	1.02 ± 0.18	0.18	0.44
	**SUVmean**	0.67 ± 0.12	0.69 ± 0.17	0.69 ± 0.13	0.26	0.87

* statistically significant.

**Table 4 diagnostics-13-01617-t004:** Semi-quantitative analysis of the abdominal aorta.

Semi-Quantitative Variable		PET1	PET2	PET3	PET1 vs. PET2 (*p*-Value)	PET2 vs. PET3 (*p*-Value)
**Abdominal** **aorta SUV**	**SUVmax**	2.66 ± 0.47	2.64 ± 0.49	2.60 ± 0.52	0.63	0.39
	**SUVmean**	1.71 ± 0.29	1.70 ± 0.26	1.68 ± 0.29	0.78	0.47
**Abdominal aorta SUV/Liver SUVmax**	**SUVmax**	0.82 ± 0.15	0.83 ± 0.13	0.81 ± 0.13	0.43	0.05
	**SUVmean**	0.52 ± 0.83	0.54 ± 0.81	0.52 ± 0.08	0.14	0.05
**Abdominal aorta SUV/Blood pool SUVmax**	**SUVmax**	1.28 ± 0.26	1.27 ± 0.23	1.29 ± 0.29	0.68	0.52
	**SUVmean**	0.82 ± 0.14	0.82 ± 0.13	0.83 ± 0.15	0.94	0.46

## Data Availability

The datasets used and/or analysed during the current study are available from the corresponding author on reasonable request.

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
