# Peer review of "Impact of [18F]FDG PET/CT in the Assessment of Immunotherapy-Induced Arterial Wall Inflammation in Melanoma Patients Receiving Immune Checkpoint Inhibitors"

_diagnostics, 2023, doi:10.3390/diagnostics13091617_

Round 1
Reviewer 1 Report
Thank you for allowing me to review this work regarding the impact of [18F]FDG PET/CT in assessing immunotherapy-induced arterial wall inflammation in melanoma patients receiving immune checkpoint inhibitors (ICIs).
The article is well-written, and the topic is intriguing.
I have just some minor suggestions for the authors:
1) In the introduction section, I suggest revising the first paragraph. Indeed melanoma is always malignant, so I believe it is incorrect to call it "malignant melanoma". Moreover, melanoma can originate not only from the skin. Please clarify this aspect in the revised manuscript.
It could be helpful to define clearly the stage (according to AJCC classification) where surgery is the gold standard.
2) I suggest clarifying the clinical exclusion criteria in the material and method section. For example, were all patients affected by primary cutaneous melanoma? Were some patients treated with target therapy before ICIs?
3) Results section: line 201 replace "bolus pemphigoid" with "bullous pemphigoid". In line 224, please clarify why ten patients were receiving glucocorticoid treatment. Were they affected by brain metastases?
4) Conclusion: the authors clearly state the study's limitations and retrospective nature. [18F]FDG PET/CT seems a promising method in melanoma patients management. However, more studies are needed to assess its role in detecting patients' risk.
Author Response
Thank you for allowing me to review this work regarding the impact of [18F]FDG PET/CT in assessing immunotherapy-induced arterial wall inflammation in melanoma patients receiving immune checkpoint inhibitors (ICIs).
The article is well-written, and the topic is intriguing.
I have just some minor suggestions for the authors:
1) In the introduction section, I suggest revising the first paragraph. Indeed melanoma is always malignant, so I believe it is incorrect to call it "malignant melanoma". Moreover, melanoma can originate not only from the skin. Please clarify this aspect in the revised manuscript.
It could be helpful to define clearly the stage (according to AJCC classification) where surgery is the gold standard.
A: Thanks for the suggestions. We changed malignant melanoma into melanoma and reworded the origin to be correct. (Lines 49-50)
Also, we defined the stage needing surgery based on the latest guideline published in 2022. (Lines 52-53)
2) I suggest clarifying the clinical exclusion criteria in the material and method section. For example, were all patients affected by primary cutaneous melanoma? Were some patients treated with target therapy before ICIs?
A: Thanks for your comment. As you said, all patients had cutaneous melanoma. We added this to the methods. (Line 94) Also, we added that systemic therapy was their first-line treatment, and in the exclusion criteria, it was noted that they did not receive any other treatment previously. Hope it is clarified now. (Lines 99-100)
3) Results section: line 201 replace "bolus pemphigoid" with "bullous pemphigoid". In line 224, please clarify why ten patients were receiving glucocorticoid treatment. Were they affected by brain metastases?
A: Thanks for your attention. Done. (Lines 187, 204, 284) Also, the issue of the glucocorticoid treatment was addressed. (Lines 228-229)
4) Conclusion: the authors clearly state the study's limitations and retrospective nature. [18F]FDG PET/CT seems a promising method in melanoma patients management. However, more studies are needed to assess its role in detecting patients' risk.
A: Thanks for your kindness. We also changed the conclusion based on your valuable suggestion. (Lines 340-342)
Reviewer 2 Report
Ranjbar S. and colleagues, in their article titled "Impact of [18F]FDG PET/CT in the assessment of immunotherapy-induced arterial wall inflammation in melanoma patients receiving immune checkpoint inhibitors" examined the role of [18F]FDG PET/CT in the early detection of arterial wall inflammation in patients receiving ICI for the treatment of advanced melanoma. The conceptualization of the article is interesting, the methods well explained, but the conclusions, in my opinion, are not strongly supported by the results. In particular, the analysis of PET/CT imaging results - both qualitatively (score 2) and quantitatively (analysis of SUVmax and SUVmean) - is fundamentally negative from my point of view. However, I suppose the authors want to emphasize a trend toward immunotherapy-induced uptake in the arterial wall, as demonstrated by statistical analysis. Moreover, the results were well discussed and the article overall interesting and innovative.
I have only a few minor suggestions:
- format the entire text
- add a PET image as an example, if possible
- check line 141 of [18F]FDG
Author Response
Ranjbar S. and colleagues, in their article titled "Impact of [18F]FDG PET/CT in the assessment of immunotherapy-induced arterial wall inflammation in melanoma patients receiving immune checkpoint inhibitors" examined the role of [18F]FDG PET/CT in the early detection of arterial wall inflammation in patients receiving ICI for the treatment of advanced melanoma. The conceptualization of the article is interesting, the methods well explained, but the conclusions, in my opinion, are not strongly supported by the results. In particular, the analysis of PET/CT imaging results - both qualitatively (score 2) and quantitatively (analysis of SUVmax and SUVmean) - is fundamentally negative from my point of view. However, I suppose the authors want to emphasize a trend toward immunotherapy-induced uptake in the arterial wall, as demonstrated by statistical analysis. Moreover, the results were well discussed and the article overall interesting and innovative.
A: Thanks for your critical comment. We reworded the conclusion to be more clarified, as you assumed correctly. (Lines 336-337)
I have only a few minor suggestions:
- format the entire text
A: We are afraid that we did not understand this comment clearly. If it is not resolved in the revised version, we kindly ask the reviewer to clarify this comment. Thanks in advance.
- add a PET image as an example, if possible
A: Thanks for your suggestion. We added a Figure based on your precious comment.
- check line 141 of [18F]FDG
A: Thanks for your attention. It is corrected now.